# BrainAlign: Leveraging EEG Foundation Models for Symmetric, Interpretable Alignment with Visual Representations

## Abstract

Custom electroencephalography (EEG) encoders trained on limited, task-specific data have restricted ability to learn generalizable, brain-like representations. We propose a representation-first alternative, leveraging a large-scale pretrained EEG foundation model (CBraMod) to learn brain-aligned representations. We introduce BrainAlign, a contrastive learning framework that uses a brain-inspired projection network to align EEG features with those from image encoders. On the challenging 200-way zero-shot visual object classification task, BrainAlign, when paired with a CORNet-S encoder, achieves a top-1 accuracy of 14.2% and a top-5 accuracy of 37.9% for EEG-to-image retrieval, performing competitively to prior baselines while reducing training time by 70%. This computational efficiency is particularly crucial for developing the subject-specific models vital for practical EEG decoding. Additionally, the framework learns a highly symmetric alignment, achieving a 23.2% top-1 and 54.7% top-5 accuracy in the reverse image-to-EEG retrieval task. We observe a time-averaged RSA correlation (r = 0.365) with the neuro-inspired CORNet-S model, consistent with a moderately high degree of representational similarity. A post-hoc CCA-INLP analysis isolates a subject-agnostic subspace and, together with a semantic similarity evaluation, shows meaningful category structure yet residual cross-subject variability. Collectively, these results in performance, efficiency, and biological plausibility provide support for our representation-first approach. The resulting robust and symmetric representations can potentially be applicable to demanding downstream applications such as object classification, high-fidelity image decoding directly from brain activity, and real-time object disambiguation.

## 1 Introduction

Aligning neural activity with representations from computational models is a fundamental approach to understanding the principles of brain function. This endeavor not only advances our basic scientific knowledge but also holds immense potential for transformative applications, particularly in developing next-generation Brain-Computer Interfaces (BCIs) for clinical and consumer use (Song et al., 2025; Liu et al., 2025). Among non-invasive neuroimaging methods, electroencephalography (EEG) offers high temporal resolution which captures neural dynamics at the millisecond scale, aligning with the rapid nature of visual processing, while its portability and low cost make it ideal for practical, real-world applications outside of laboratory settings (Song et al., 2025; Trafton, n.d.). In contrast, modalities such as fMRI provide superior spatial resolution but different practical trade-offs (e.g., cost and immobility). (Sharon et al., 2007).

Historically, decoding was hindered by low signal-to-noise ratios and by potential temporal confounds in some block-design paradigms. (Song et al., 2025; Xu et al., 2021). The field has since shifted toward more robust methodologies, with the Rapid Serial Visual Presentation (RSVP) paradigm and large-scale datasets like THINGS-EEG2 enabling the study of neural responses to thousands of natural images (Gifford et al., 2022). This evolution led to self-supervised contrastive learning emerging as the dominant approach for aligning the high-dimensional space of EEG signals with rich visual representations (Song et al., 2025). However, a critical limitation pervades these modern methods: they almost exclusively rely on custom EEG encoders trained from scratch on a single

alignment task. This methodology can be constrained, as an encoder optimized solely for one task is unlikely to learn the generalizable, brain-like neural codes that capture the full richness of brain activity. To overcome this, we propose an alternative "representation-first" approach that leverages the power of EEG foundation models (Berto, n.d.). These models, pre-trained on massive and diverse neural datasets, learn universal and robust representations that serve as a superior starting point. By fine-tuning from this rich representational base, we can learn alignments that are more data-efficient, performant, and, importantly, more likely to be biologically plausible (Jiang et al., 2024; Wang et al., 2024).

To rigorously evaluate the quality of the learned representations, we utilize the 200-way zero-shot visual object classification task. This task serves as a challenging benchmark for two reasons: First, its zero-shot nature directly tests the model's ability to generalize to unseen semantic concepts, a key indicator of a robustly learned representation space. Second, it is an established evaluation paradigm within the BCI and neuro-AI communities (Du et al., 2023; Song et al., 2023; 2025), allowing for direct comparison with prior state-of-the-art methods. Success on this task, therefore, is not an end in itself, but a commonly-used proxy for the quality and generalizability of the underlying brain-visual alignment.

To implement this representation-first approach, we introduce BrainAlign, a framework designed for the symmetric and interpretable alignment of EEG and visual representations. While leveraging a foundation model addresses the primary challenge of learning robust neural codes, our framework is also designed to investigate several other critical gaps in existing research. First, unlike architecturally asymmetric models, BrainAlign is designed to be bidirectional, capturing the reciprocal nature of information processing in the brain (Zhang et al., 2025; Qiao et al., 2019). Second, we move beyond "black box" models by incorporating methods that enhance mechanistic interpretability, allowing us to use the model as a scientific instrument. Finally, we address the open question of which visual feature space best aligns with EEG signals. By systematically comparing a purely hierarchical model (ResNet (He et al., 2016)), a brain-inspired recurrent model (CORNet-S (Kubilius et al., 2019)), and a vision-language model (CLIP (Lu & Wang, 2025)), we can probe the nature of the optimal visual-neural alignment. Beyond the core BrainAlign results, we ask which part of the learned EEG–image representation is subject-agnostic (stimulus-driven) versus subject-dependent (identity) and how the representation is arranged with respect to higher-level categories. We therefore include a simple post-hoc linear analysis: (i) align EEG and image embeddings with CCA and (ii) excise linearly decodable subject information via iterative nullspace projection (INLP), contrasting with a mean-subspace removal baseline. Additionally, we quantify semantic structure using retrieval-style and representational-similarity metrics (MRR, NDCG, AUC, within-between margins).

This paper introduces a framework for visual object classification from EEG that directly addresses the aforementioned gaps. Our contributions can be summarized as follows: (a) we introduce BrainAlign, a framework that leverages a pretrained EEG foundation model (CBraMod (Wang et al., 2024)) for a representation-first approach to aligning EEG and visual features; (b) we systematically compare the alignment of EEG representations with three neuroscientifically motivated visual backbones: ResNet-50 (He et al., 2016), CORNet-S (Kubilius et al., 2019), and CLIP (Radford et al., 2021); (c) we demonstrate the bidirectional symmetry of the learned representation space, enabling both decoding and encoding applications; (d) we assess interpretability by visualizing learned importance weights corresponding to distinct brain regions; (e) we analyze the quality of the shared representation space through its intrinsic information content and downstream task performance.

## 2 RELATED WORK

**Aligning neural and computational models.** The effort to map visual representations in the brain has progressed from early fMRI studies, which established that object categories could be decoded from cortical activity (Song et al., 2025), to modern electrophysiological methods like EEG. The high temporal resolution of EEG is better suited to capture the rapid dynamics of visual perception (Berto, n.d.). A significant methodological advance was the adoption of the Rapid Serial Visual Presentation (RSVP) paradigm, which, combined with large-scale datasets, enabled the field to move beyond simple classification to ambitious zero-shot decoding tasks using deep learning (Gifford et al., 2022; Jiao et al., 2019). This research now largely falls under the broader goal of integrative benchmarking, where computational models are quantitatively evaluated on their ability to predict

neural and behavioral data, a practice formalized by platforms like Brain-Score (Schrimpf et al., 2020).

**Contrastive learning for EEG-vision alignment.** The current state-of-the-art for aligning EEG signals with visual features is self-supervised contrastive learning (Liu et al., 2021). The pioneering NICE framework demonstrated that a contrastive loss could effectively map EEG and image embeddings (e.g., from CLIP) into a shared space for zero-shot recognition (Song et al., 2023). While language-guided extensions like NICE++ have shown performance gains by using textual descriptions to refine the alignment (Song et al., 2025), they do so by introducing a third modality (language). As our work is focused on the fundamental principles of direct EEG-vision alignment, we compare against uni-modal visual encoders. Subsequent work has introduced sophisticated refinements to address challenges such as the "modality gap". For instance, BraVL uses a multimodal VAE to learn a unified latent space (Du et al., 2023), VE-SDN introduces a semantic decoupling module to align only the shared information (Chen et al., 2024), and others leverage guidance from large language models to refine the alignment (Song et al., 2025). A common thread, however, unites these advanced methods: they all train their EEG encoders from scratch for a specific alignment task. This approach is fundamentally limited, as the encoders must simultaneously learn basic neural feature extraction and high-level semantic alignment, a challenge that our work directly addresses.

**EEG foundation models.** These models are pre-trained on massive and diverse EEG corpora, such as the TUH-EEG dataset (Obeid & Picone, 2016), to learn universal, robust, and generalizable representations of brain activity. Architectures like BENDR (Kostas et al., 2021) and LaBraM (Jiang et al., 2024) established the viability of this approach. We employ CBraMod (Wang et al., 2024), a state-of-the-art foundation model whose criss-cross transformer architecture is uniquely suited to capturing the spatio-temporal dynamics of EEG. By starting with these rich, pre-trained representations, we reframe the problem from one of end-to-end training to one of targeted fine-tuning. This aligns with a broader movement in computational neuroscience away from purely predictive "black box" models and toward models that are mechanistically interpretable (Krakauer et al., 2017). The goal is to build transparent, falsifiable models of neural computation, where the internal workings can be causally linked to behavior and brain activity. Our representation-first approach, grounded in a powerful foundation model, is a significant step in this direction.

## 3 METHOD

The methodology of this study is designed to validate our central thesis: that leveraging a pre-trained EEG foundation model provides a more robust and biologically plausible pathway to learning brain-aligned representations than training task-specific encoders from scratch. To this end, we introduce BrainAlign, a framework designed for the symmetric and interpretable alignment of EEG and visual features. Our experimental design adheres to a subject-dependent paradigm. This choice is rooted in the principle of biological plausibility; as each human brain possesses unique functional characteristics, developing subject-specific models is essential for capturing genuine neural representations, rather than learning a non-representative 'average' brain model. In this section, we will detail the architecture of the BrainAlign framework (refer Figure 1), the rationale behind its components, and the contrastive learning procedure used for training. We also briefly discuss the procedures used to perform the post-hoc analyses.

### 3.1 BRAINALIGN ARCHITECTURE

The BrainAlign framework consists of two parallel processing streams—an EEG branch and an image branch—that learn to project their respective outputs into a shared representation space. The EEG branch is designed to address the fundamental limitations of conventional approaches that train encoders from scratch. Such methods are not only computationally expensive (e.g., up to 200 epochs (Song et al., 2023)) but also risk learning brittle, task-specific representations, as they must learn low-level features and high-level alignment simultaneously. Our framework circumvents this by utilizing a pre-trained EEG foundation model, CBraMod (Wang et al., 2024), as the encoder. By starting with the rich, general-purpose representations learned from diverse datasets (Jiang et al., 2024; Kostas et al., 2021; Obeid & Picone, 2016), our model can achieve high performance with substantially less fine-tuning. Following this encoder, we introduce a custom projection network

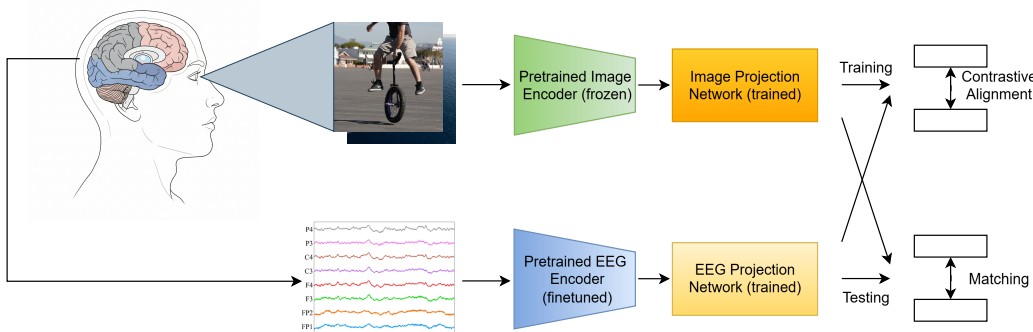

Figure 1: The BrainAlign framework for EEG foundation model-based object classification. The framework relies on powerful pretrained EEG and image encoders, and while fine-tuning the EEG encoder, trains the projection networks using contrastive learning to align the representation spaces from both branches. Testing is done by matching EEG branch representations with pre-obtained image branch templates for test images.

designed with strong neuroscientific priors. The architecture adopts a multi-stream design that segregates channels into functionally distinct groups (occipital, parietal, temporal, and global) and integrates them via a learnable gating mechanism, yielding a functionally grounded and interpretable embedding. The detailed mathematical formulation of this regional aggregation process is provided in Appendix A.

A central scientific question of this study is what kind of computational visual feature space aligns most effectively with neural representations. To investigate this, the image branch of our framework is designed to be modular. We systematically compare three distinct, neuroscientifically motivated image encoders, each representing a different hypothesis about visual processing: a hierarchical feedforward model (ResNet-50), a brain-inspired recurrent model (CORNet-S), and a multimodal vision-language model (CLIP). This comparative experiment is therefore designed not simply to find the best-performing model, but to use alignment performance as evidence to adjudicate between these competing computational theories of visual representation. A detailed description of each of these encoders is available in Appendix A. Following the selected encoder, a simple 2-layer MLP with GeLU activation serves as a projection network to map the image features into the shared representation space.

## 3.2 CONTRASTIVE LEARNING

The core of the training process is to align EEG and image features in a shared embedding space. This is achieved using a symmetric contrastive loss function, similar to the one introduced in CLIP. The symmetric nature of this loss is critical, as it encourages the learned latent space to be bidirectionally informative. This ensures that an EEG representation can be used to identify its corresponding image (decoding) and, equally, that an image representation can identify its EEG counterpart (encoding), a property essential for building models that reflect the brain's reciprocal processing pathways.

Given a mini-batch of $N$ paired EEG and image samples, we first extract their respective feature vectors, $\boldsymbol{f}_e$ and $\boldsymbol{f}_i$, using the EEG and image encoders. These features are then projected into a shared embedding space of dimension $D$ by projection heads $P_{eeg}$ and $P_{img}$.

The projected features for the $k$-th sample are denoted as $\boldsymbol{z}_e^{(k)} = P_{eeg}(\boldsymbol{f}_e^{(k)})$ and $\boldsymbol{z}_i^{(k)} = P_{img}(\boldsymbol{f}_i^{(k)})$. These features are L2-normalized:

$$\hat{\boldsymbol{z}}_e^{(k)} = \frac{\boldsymbol{z}_e^{(k)}}{\left\| \boldsymbol{z}_e^{(k)} \right\|_2} \quad \text{and} \quad \hat{\boldsymbol{z}}_i^{(k)} = \frac{\boldsymbol{z}_i^{(k)}}{\left\| \boldsymbol{z}_i^{(k)} \right\|_2}$$

The similarity between the $j$-th EEG feature vector and the $k$-th image feature vector in the batch is calculated as the cosine similarity (dot product of normalized vectors), scaled by a learnable temperature parameter $\tau$:

$$s_{jk} = \tau \cdot \left\langle \hat{\boldsymbol{z}}_e^{(j)}, \hat{\boldsymbol{z}}_i^{(k)} \right\rangle$$

The objective is to maximize the similarity of corresponding pairs (where $j = k$) while minimizing it for all other non-corresponding pairs within the batch. This is framed as a classification problem using the cross-entropy loss. The loss is calculated symmetrically for both EEG-to-image and image-to-EEG directions.

The loss for predicting the correct image pairing for a given EEG signal is:

$$\mathcal{L}_{\text{eeg}} = -\frac{1}{N} \sum_{j=1}^{N} \log \frac{\exp\left(s_{jj}\right)}{\sum_{k=1}^{N} \exp\left(s_{jk}\right)}$$

Similarly, the loss for predicting the correct EEG pairing for a given image is:

$$\mathcal{L}_{\text{img}} = -\frac{1}{N} \sum_{j=1}^{N} \log \frac{\exp\left(s_{jj}\right)}{\sum_{k=1}^{N} \exp\left(s_{kj}\right)}$$

The final training objective is the average of these two losses:

$$\mathcal{L}_{\text{total}} = \frac{\mathcal{L}_{\text{eeg}} + \mathcal{L}_{\text{img}}}{2}$$

## 4 EXPERIMENTAL SETUP AND RESULTS

### 4.1 DATASET

We used the THINGS-EEG2 dataset (Gifford et al., 2022), which contains EEG responses from 10 subjects viewing natural images in a rapid serial visual presentation (RSVP) paradigm, making it ideal for studying object recognition. Among the few high-quality EEG-image datasets relevant for this task, this specific dataset was chosen for its scale and established validity. We followed standard preprocessing procedures and, unlike prior work that used a subset of channels, we retained all 63 recording channels to provide a more complete representation of the distributed neural activity for our model. A detailed description of the dataset, our full preprocessing pipeline, and a data quality analysis that validates the use of all channels, are provided in Appendix B.

### 4.2 EVALUATION FRAMEWORK AND RESULTS

Our experimental investigation centered on two key questions, evaluated on a subject-dependent basis to account for inter-subject variability (Saha & Baumert, 2020). First, to test our central hypothesis, we compared two training strategies for the CBraMod encoder: fine-tuning the pre-trained weights versus keeping them frozen. Second, to investigate the nature of the optimal visual feature space, we paired each EEG strategy with the three visual backbones (ResNet-50, CORNet-S, and CLIP). This resulted in six model configurations per subject, which were evaluated on the bidirectional 200-way zero-shot classification task (chance-level accuracy: 0.5%). For a deeper, qualitative assessment of the learned representations, we also designed a series of targeted representational analyses (e.g., representational similarity analysis (RSA), time-resolved encoding). A detailed description of each of these representational analysis methods is provided in Appendix D.

The performance of our six model configurations was evaluated and compared against the NICE, NICE-GA, and BraVL frameworks (Song et al., 2023; Du et al., 2023). In this work, we focus our primary analysis on top-1 accuracy, as it serves as the most stringent metric for evaluating the quality and "brain-alikeness" of the learned representations. Unlike top-5 accuracy, which allows for a wider

Table 1: A comparison of different model performances (top-1 accuracies) across 10 subjects for the EEG-to-image 200-way zero-shot classification task

| Method | S1 | S2 | S3 | S4 | S5 | S6 | S7 | S8 | S9 | S10 | Mean | SD |
|---|---|---|---|---|---|---|---|---|---|---|---|---|
| BraVL (Du et al., 2023) | 6.1 | 4.9 | 5.6 | 5.0 | 4.0 | 6.0 | 6.5 | 8.8 | 4.3 | 7.0 | 5.8 | 1.4 |
| NICE (Song et al., 2023) | 12.3 | 10.4 | 13.1 | 16.4 | 8.0 | 14.1 | 15.2 | 20.0 | 13.3 | 14.9 | 13.8 | 3.3 |
| NICE-GA (Song et al., 2023) | 15.2 | 13.9 | 14.7 | 17.6 | 9.0 | 16.4 | 14.9 | 20.3 | 14.1 | 19.6 | 15.6 | 3.2 |
| CBraMod (fine-tuned) + CLIP | 14.5 | 9.5 | 14.0 | 11.5 | 10.0 | 19.0 | 11.5 | 16.5 | 13.5 | 17.0 | 13.7 | 3.1 |
| CBraMod (fine-tuned) + ResNet-50 | 12.0 | 12.0 | 12.0 | 9.5 | 9.0 | 21.5 | 12.0 | 16.0 | 10.0 | 18.5 | 13.2 | 4.1 |
| CBraMod (fine-tuned) + CORNet-S | 11.5 | 13.0 | 13.5 | 16.0 | 10.0 | 20.5 | 14.5 | 14.0 | 12.5 | 16.5 | 14.2 | 2.9 |
| CBraMod (frozen) + CLIP | 2.5 | 5.0 | 7.0 | 7.5 | 2.5 | 6.5 | 5.0 | 7.0 | 4.5 | 10.0 | 5.7 | 2.3 |
| CBraMod (frozen) + ResNet-50 | 5.0 | 5.5 | 6.5 | 4.5 | 6.0 | 9.0 | 5.0 | 10.0 | 2.5 | 6.5 | 6.0 | 2.2 |
| CBraMod (frozen) + CORNet-S | 4.0 | 6.5 | 7.0 | 5.5 | 6.0 | 8.5 | 5.5 | 7.5 | 2.5 | 9.0 | 6.2 | 2.0 |

Table 2: A comparison of different model performances (top-1 accuracies) across 10 subjects for the image-to-EEG 200-way zero-shot classification task

| Method | S1 | S2 | S3 | S4 | S5 | S6 | S7 | S8 | S9 | S10 | Mean | SD |
|---|---|---|---|---|---|---|---|---|---|---|---|---|
| CBraMod (fine-tuned) + CLIP | 23.0 | 17.0 | 16.0 | 20.0 | 17.5 | 23.0 | 19.0 | 26.5 | 18.5 | 30.5 | 21.1 | 4.6 |
| CBraMod (fine-tuned) + ResNet-50 | 17.0 | 26.5 | 19.5 | 22.5 | 21.0 | 29.0 | 15.5 | 24.5 | 13.5 | 29.0 | 21.8 | 5.5 |
| CBraMod (fine-tuned) + CORNet-S | 17.0 | 25.5 | 21.5 | 25.0 | 18.0 | 33.5 | 23.0 | 27.0 | 16.0 | 26.0 | 23.2 | 5.3 |
| CBraMod (frozen) + CLIP | 4.5 | 7.5 | 9.5 | 11.5 | 8.5 | 10.5 | 5.5 | 13.5 | 2.5 | 11.0 | 8.4 | 3.4 |
| CBraMod (frozen) + ResNet-50 | 6.0 | 10.5 | 6.5 | 12.0 | 10.5 | 12.0 | 5.5 | 13.0 | 5.5 | 8.0 | 8.9 | 3.0 |
| CBraMod (frozen) + CORNet-S | 3.5 | 10.0 | 9.5 | 7.0 | 9.0 | 12.0 | 4.5 | 12.5 | 4.0 | 13.0 | 8.5 | 3.6 |

margin of error, top-1 accuracy directly probes the model's ability to select the single correct item from 200 distinct choices. This provides a direct measure of the representation's discriminative power—its ability to distinguish between fine-grained concepts from neural data, which is a key characteristic of the brain's own highly specific and efficient visual processing system. The mean top-1 accuracies across all subjects are presented in Table 1 and Table 2; for completeness, top-5 accuracies are provided in Appendix G.

Our primary finding supports the central hypothesis of this work: leveraging a pre-trained foundation model as an inductive bias via fine-tuning is superior to using it as a static feature extractor. As shown in Tables 1 and 2, all fine-tuned models outperformed their frozen-backbone counterparts. This large and statistically significant improvement in top-1 accuracy ($p < 0.01$, Wilcoxon signed-rank test) demonstrates that the fine-tuning process is important for adapting the foundation model's general-purpose features into a highly discriminative semantic space, one that is better suited for the specific task of visual object recognition from EEG. This result supports our representation-first approach.

Having established the importance of fine-tuning, we next investigated which visual feature space aligns best with the adapted EEG representations. Among the fine-tuned models, the configuration using the brain-inspired recurrent CORNet-S encoder achieved the highest average top-1 accuracy in both EEG-to-image (14.2%) and image-to-EEG (23.2%) directions. This suggests that its representations, shaped by recurrent connections designed to mimic the primate ventral stream, provide a more suitable target space for alignment with neural data. However, differences among visual backbones were not statistically significant ($p > 0.05$), so we view this as a tentative trend rather than conclusive evidence for any specific architecture.

Our best-performing model (CBraMod fine-tuned + CORNet-S) is highly competitive with current state-of-the-art methods, significantly outperforming BraVL (5.8%) and the base NICE (13.8%) frameworks, and achieving an accuracy comparable to the more complex NICE-GA model (15.6%). Crucially, this performance is achieved with marked computational efficiency. All fine-tuned models converged within 60 epochs, a 70% reduction in training time. This reduction could make subject-specific training more practical. As a new model must be trained for each new subject, a significant reduction in training time directly translates to lower computational costs and a greater capacity to apply the framework to larger participant cohorts.

## 4.3 MODEL INTERPRETABILITY AND REPRESENTATIONAL PLAUSIBILITY

To assess model interpretability, we visualized the regional importance weights learned by the EEG projection network as a topographical map (Figure 2). The visualization shows that the model

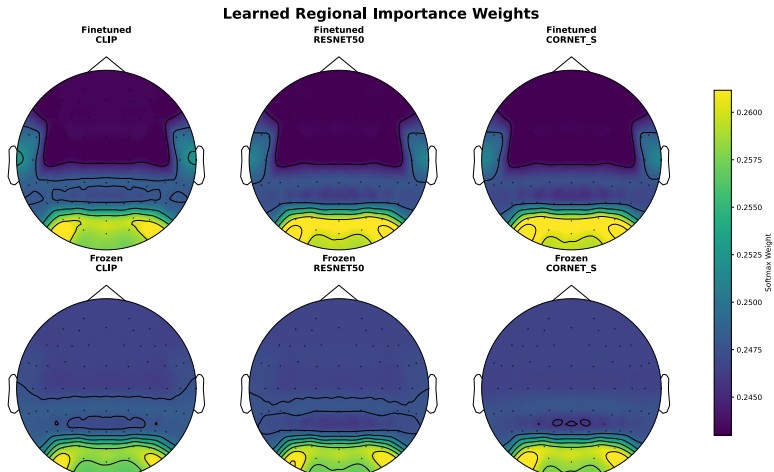

Figure 2: Topographical map of brain region importance weights learned by the EEG projection network.

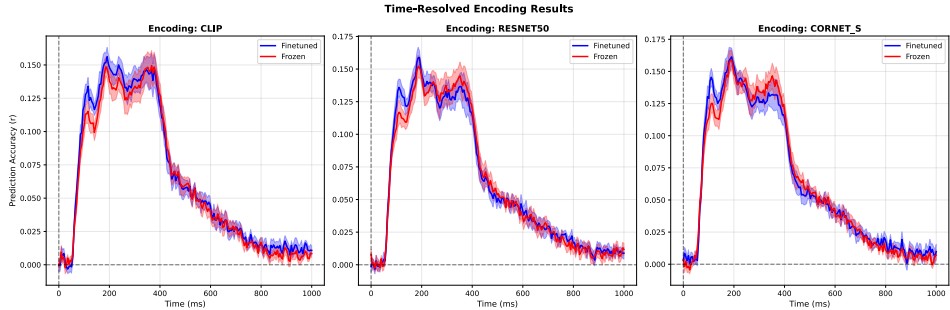

Figure 3: Prediction accuracy of raw EEG signals from image representations using time-resolved encoding models.

consistently assigned higher weights to occipital, parieto-occipital, and inferior temporal channels compared to frontal channels. This learned weight distribution is consistent with the known functional anatomy of the ventral visual pathway, providing evidence for the biological plausibility of the model. Furthermore, the fine-tuned models learned a weight distribution that more closely resembled this neuroscientific prior compared to the frozen-backbone models. This observation provides a potential mechanistic explanation for the performance gap reported in Section 4.2: the fine-tuning process not only adapts the feature space but also increases emphasis on occipito-temporal channels, thus hinting towards enhanced biological plausibility. Although more targeted analyses would be needed to firmly establish a causal neuroanatomically correct attention policy. The superior performance of the fine-tuned models is therefore not just a numerical result, but a potential consequence of learning a more biologically plausible processing strategy.

To provide deeper evidence for the quality of the learned representations beyond classification accuracy, we conducted a series of representational analyses (see Appendix E for full details). These analyses confirmed three key points. First, time-resolved encoding showed that our aligned representations captured significant, dynamically evolving neural information, mirroring the known temporal progression of the visual (Figure 3). Second, Representational Similarity Analysis (RSA) revealed that the geometry of the space learned by the fine-tuned models had a significantly higher correlation with the brain's own representational geometry compared to the frozen models (Appendix Figure 7). Third, high accuracy on cross-modal retrieval tasks confirmed that the space is robustly bidirectional. Taken together, these results provide converging evidence that the performance gains from our foundation model framework are rooted in its ability to learn a shared latent space that is more structurally and dynamically aligned with the brain's internal representations.

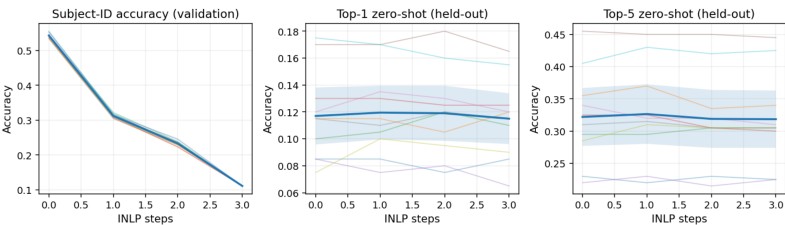

Figure 4: Subject-ID, top-1 and top-5 accuracy against number of INLP steps. The subject-ID accuracies fall while top-1 and top-5 image classification accuracies remain stable.

Table 3: CCA canonical spectrum (mean $\pm$ s.e., over 10 folds). High, tight values indicate a robust shared latent.

| CC1 | CC2 | CC3 | CC4 | CC5 | CC6 | CC7 | CC8 | CC9 | CC10 |
|---|---|---|---|---|---|---|---|---|---|
| 0.863$\pm$ 0.000 | 0.859 $\pm$ 0.001 | 0.854 $\pm$ 0.001 | 0.852 $\pm$ 0.000 | 0.850 $\pm$ 0.000 | 0.850 $\pm$ 0.000 | 0.847 $\pm$ 0.001 | 0.844 $\pm$ 0.001 | 0.835 $\pm$ 0.001 | 0.815 $\pm$ 0.001 |

### 4.4 ADDITIONAL REPRESENTATIONAL ANALYSES

To further probe the learned representations, we performed two additional analyses using a leave-one-subject-out (LOSO) evaluation. First, we aimed to disentangle subject-agnostic (stimulus-driven) information from subject-dependent (identity) features. Second, we quantified the semantic structure of the shared representation space. A full description of the methods and detailed results are in Appendix F.

Using a CCA-INLP pipeline, we found that we can remove linearly decodable subject identity information from the EEG features without degrading zero-shot recognition performance. As shown in Figure 4 and Table 4, INLP reduces the subject-ID leakage to chance level while top-1 and top-5 accuracies remain stable. The high canonical correlations (Table 3) indicate a robust shared latent space across subjects. This suggests that subject-specific information occupies a compact linear subspace that can be excised to isolate a predominantly subject-agnostic, stimulus-driven representation.

Our analysis of the subject-averaged EEG-to-image similarity matrix reveals a non-trivial semantic organization. Retrieval metrics (Table 5) show that correct and same-category items are ranked highly. The positive within-between category margins (Appendix Table 7) and the qualitative visualizations (Figure 5) further confirm that the shared space captures meaningful semantic relationships, with clear categorical structure for concepts like weapons and plants, while others, such as vegetables, show less separation. However, modest global AUC and centroid consistency suggest that some cross-subject idiosyncrasies remain in the shared geometric space.

## 5 CONCLUSION AND FUTURE WORK

In this work, we find that leveraging pre-trained EEG foundation models via fine-tuning is associated with improved efficiency and higher alignment metrics in our setting for aligning neural and artificial visual representations. Our BrainAlign framework achieves competitive performance on the challenging 200-way zero-shot classification benchmark while drastically reducing the required training time by 70%. Importantly, this quantitative performance is underpinned by qualitative evidence of greater

Table 4: LOSO summary for identity removal. Leakage = subject-ID accuracy (on a validation split of the 9-subject pool). Zero-shot top-1 and top-5 = 200-way EEG-to-image identification on the held-out subject. INLP achieves chance leakage without hurting top-1/top-5.

| Method | Leakage start | Leakage end | Top-1 start | Top-1 end | Top-5 start | Top-5 end |
|---|---|---|---|---|---|---|
| CCA-INLP | 0.543$\pm$0.003 | 0.111$\pm$0.000 | 0.117$\pm$0.011 | 0.115$\pm$0.010 | 0.322$\pm$0.023 | 0.319$\pm$0.023 |
| Mean-subspace ($r=8$) | 0.895$\pm$0.003 | 0.288$\pm$0.003 | 0.143$\pm$0.010 | 0.140$\pm$0.009 | 0.380$\pm$0.024 | 0.381$\pm$0.025 |

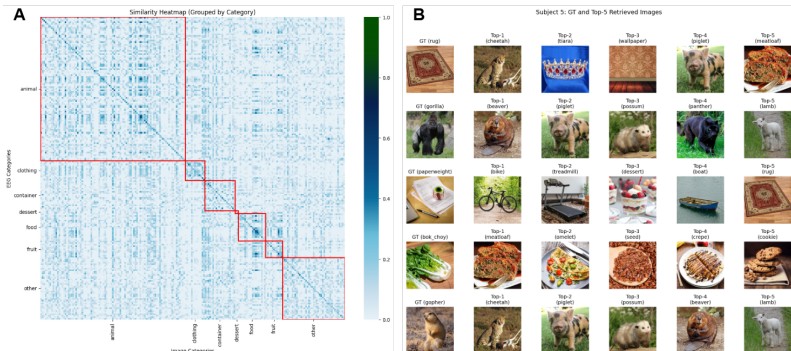

Figure 5: Results of semantic similarity analysis. (A) Cosine similarity between EEG and image representations averaged over all 10 subjects (10 less frequent categories were grouped together as "other"). (B) Qualitative retrieval: Ground truth (GT) (col. 1) and top-1...top-5 for three queries (rows). Subject 5 shown.

Table 5: Semantic similarity (averaged across 10 subjects). MRR and NDCG@10 capture ranking quality beyond exact match; AUC is threshold-free category separability; within–between $\Delta$ and $d$ quantify block coherence; block-energy ratio summarizes the fraction of similarity mass inside category blocks; centroid consistency measures cross-subject alignment of category geometry.

| MRR | NDCG@10 | AUC | Within-Between $\Delta$ | Cohen's d | Block energy | Centroid consistency |
|---|---|---|---|---|---|---|
| 0.563 | 0.504 | 0.543 | 0.0197 | 0.187 | 0.480 | 0.375 $\pm$ 0.021 |

neuroscientific validity: interpretability analyses reveal that our fine-tuned model learns a biologically plausible attentional policy, while representational similarity analyses confirm that its learned geometry is more congruent with the brain's own. The last set of representational analyses show that a compact linear subject-agnostic subspace supports zero-shot recognition while subject information can be removed to chance post hoc, and that mean/second-order shifts alone cannot explain identity leakage. The residual cross-subject variance in semantic structure motivates training-time invariance (e.g., domain-adversarial objectives) and cross-subject alignment (e.g., hyperalignment-style mappings) as complementary future work. These findings collectively establish the "representation-first" approach as a robust and scientifically informative path forward, which has the potential to enable the development of more sophisticated BCIs and more transparent computational models of brain function.

**Limitations.** All results are based on subject-dependent models, and therefore, cross-subject generalization remains to be explored yet. The 200-way zero-shot classification task, while a good and commonly-used proxy for measuring quality of alignment, leaves actual downstream task performance on tasks like image reconstruction to future work. While we tried to establish interpretability in various ways, large-scale user studies are required to demonstrate the biological plausibility of the model, which is beyond the scope of this study.

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

# A    ARCHITECTURAL AND MODEL DETAILS

## A.1    EEG PROJECTION NETWORK FORMULATION

The process for deriving the aggregated EEG vector from the output of the EEG encoder, $F = \{\boldsymbol{f}_1, \boldsymbol{f}_2, \ldots, \boldsymbol{f}_C\}$, is as follows. The channels are grouped into four disjoint sets based on their location: occipital ($C_O$), parietal ($C_P$), temporal ($C_T$), and other ($C_{Other}$). For each region $R \in \{O, P, T, Other\}$, the features are first averaged:

$$\bar{\boldsymbol{f}}_R = \frac{1}{|C_R|} \sum_{c \in C_R} \boldsymbol{f}_c$$

This mean-pooled feature vector is then passed through a region-specific projection network $P_R$:

$$\boldsymbol{f}'_R = P_R(\bar{\boldsymbol{f}}_R)$$

The model learns a set of importance weights, $\boldsymbol{w} = [w_O, w_P, w_T, w_{Other}]$, which are derived from a learnable parameter parameter vector $\boldsymbol{v}$ via the softmax function:

$$\boldsymbol{w} = \text{softmax}(\boldsymbol{v})$$

Finally, the weighted features from each region are concatenated to form the final aggregated EEG feature vector, $\boldsymbol{z}_{agg}$:

$$\boldsymbol{z}_{\text{agg}} = [w_O \cdot \boldsymbol{f}'_O \oplus w_P \cdot \boldsymbol{f}'_P \oplus w_T \cdot \boldsymbol{f}'_T \oplus w_{Other} \cdot \boldsymbol{f}'_{Other}]$$

where $\oplus$ denotes the concatenation operation.

## A.2    IMAGE ENCODER DETAILS

We systematically compare three distinct image encoders, each representing a different hypothesis about visual processing.

### A.2.1    RESNET-50

This model (He et al., 2016) represents the 'hierarchical feedforward' hypothesis, where visual information is processed through a series of increasingly complex, feedforward layers. Its alignment performance serves as a baseline for a standard, highly-performant computer vision architecture.

### A.2.2    CORNET-S

This model (Kubilius et al., 2019) represents the 'brain-inspired recurrence' hypothesis. It was explicitly designed to model the primate ventral visual stream and incorporates recurrent connections, which are a key feature of the visual cortex. Its performance tests whether an architecturally more brain-like model yields better alignment.

### A.2.3 CLIP

This model (Radford et al., 2021) represents the 'semantic embedding' hypothesis. Pre-trained on image-text pairs, its representations are not purely visual but are deeply structured by language and semantics. Its performance probes whether the brain's representation of objects is more akin to a rich, multimodal semantic space than a purely visual one.

## B DATASET DETAILS

### B.1 DATASET AND PREPROCESSING

For this study, we selected the THINGS-EEG2 (Gifford et al., 2022) dataset due to its neuroscientific validity and high temporal resolution. This dataset contains EEG responses from 10 subjects viewing natural images presented using a rapid serial visual presentation (RSVP) paradigm. The RSVP protocol is designed to elicit stimulus-specific neural responses while minimizing contributions from higher-order cognitive processes, making the data suitable for training models on object recognition. The dataset comprises 82,160 trials across 16,740 unique image conditions, which map to 1,854 object classes. We adhere to the original study's split, using 1,654 classes for training and 200 classes for the zero-shot evaluation task. For the test set, one image per class was selected for the 200-way classification task. EEG data was recorded from 64 channels using an EASYCAP system, out of which 63 were recording channels and one was stimulus channel.

We followed standard EEG preprocessing steps, consistent with those applied by Song et al.. The raw data was epoched into 1000 ms trials post-stimulus onset and baseline-corrected using the mean of the 200 ms pre-stimulus period. A bandpass filter was applied to retain frequencies between 0.1 and 100 Hz. For all analyses, the data was down-sampled from 1000 Hz to 250 Hz, and multivariate noise normalization was performed to reduce correlated noise across channels. This frequency was chosen in accordance with the Nyquist-Shannon sampling theorem. All trial repetitions for each image condition were averaged to increase the signal-to-noise ratio. During training, the EEG data was further down-sampled to 200 Hz to match the input requirements of the CBraMod foundation model. For the image branch, we utilized pre-computed image representations from ResNet-50, CORNet-S, and CLIP, as provided by the original dataset creators and Song et al., to facilitate faster model training and evaluation.

### B.2 DATASET QUALITY ANALYSIS

While prior work has sometimes restricted analysis to 17 occipital and parietal channels, we retained all 63 channels for model training, similar to Song et al (Song et al., 2023). This decision is motivated by the fact that the ventral visual pathway, which is critical for object recognition, extends beyond the occipital and parietal lobes into the inferior temporal cortex (Bao et al., 2020). Including all channels allows the model to potentially capture a more complete representation of the distributed neural activity underlying visual processing. Our model architecture is designed to leverage these additional channels while enabling interpretability of region-specific contributions.

To confirm the data quality across all channels, we performed a temporal and spatial analysis of the EEG responses, as the original dataset's analyses primarily focused on a smaller subset of channels. Figure 6 displays topographical maps of the average EEG response over time. The activation patterns are consistent with established neuroscientific findings: an initial increase in activity in the occipital lobe (0-100 ms), followed by propagation to the temporal lobe, which is characteristic of feedforward processing along the ventral visual stream which includes processing along V1, V2, V3, PIT, CIT and AIT areas. This analysis suggests the suitability of the full 63-channel dataset for our task.

## C HYPERPARAMETER CHOICES

The hyperparameters used for training all models are provided in Table 6 (Song et al., 2023).

Figure 6: Topographical maps of EEG responses from one subject averaged over all training image conditions across 10 time intervals.

Table 6: Hyperparameter settings used for model training.

| Name | Value |
|---|---|
| Batch size | 1024 |
| Learning rate | 0.0002 |
| Adam $\beta_1$ | 0.5 |
| Adam $\beta_2$ | 0.999 |
| Logit scale ($\tau$) | $\log(1/0.07)$ |
| Projection dimension (EEG and Image) | 800 |
| EEG encoder embedding dimension | 800 |
| Image encoder embedding dimension (CLIP) | 784 |
| Image encoder embedding dimension (CORNet-S and ResNet-50) | 3000 |
| Dropout (all layers) | 0.2 |
| Validation split size | 740 samples |
| Training split size | 16540 samples |
| Test split size | 200 samples |

## D  REPRESENTATIONAL ANALYSIS METHODS

To gain deeper insight into the structure and biological plausibility of the shared latent space, we conducted a series of targeted representational analyses, as described below.

**Quality of neural information content**    To verify that the aligned image representations captured meaningful neural information, we performed a time-resolved encoding analysis. Using a nested cross-validated Ridge regression model, we predicted EEG signals at each time point from the static image features of the aligned space. High prediction accuracy in this analysis would indicate that the contrastive learning process successfully embedded neurally-relevant visual features into the representations, validating the image-to-EEG mapping.

**Similarity to brain's representational geometry**    To assess the biological plausibility of the learned space, we compared its internal structure to that of the brain using time-resolved Representational Similarity Analysis (RSA) (Kriegeskorte et al., 2008). We computed Representational Dissimilarity Matrices (RDMs) for the model and for the neural data at each time point. A high correlation between the model and brain RDMs over time would indicate that our framework learns a representational geometry that dynamically mirrors the brain's own processing trajectory.

**Bidirectional symmetry and alignment**    Finally, to evaluate the overall alignment and bidirectional utility of the final shared space, we conducted two analyses. First, a static RSA measured the global alignment between the final EEG and image representational geometries. Second, a cross-modal retrieval task directly tested the framework's symmetry by evaluating its ability to retrieve the correct EEG vector from its image counterpart, and vice-versa. Success in these tasks is a direct measure of how well the two modalities were fused into a coherent, symmetric representational space.

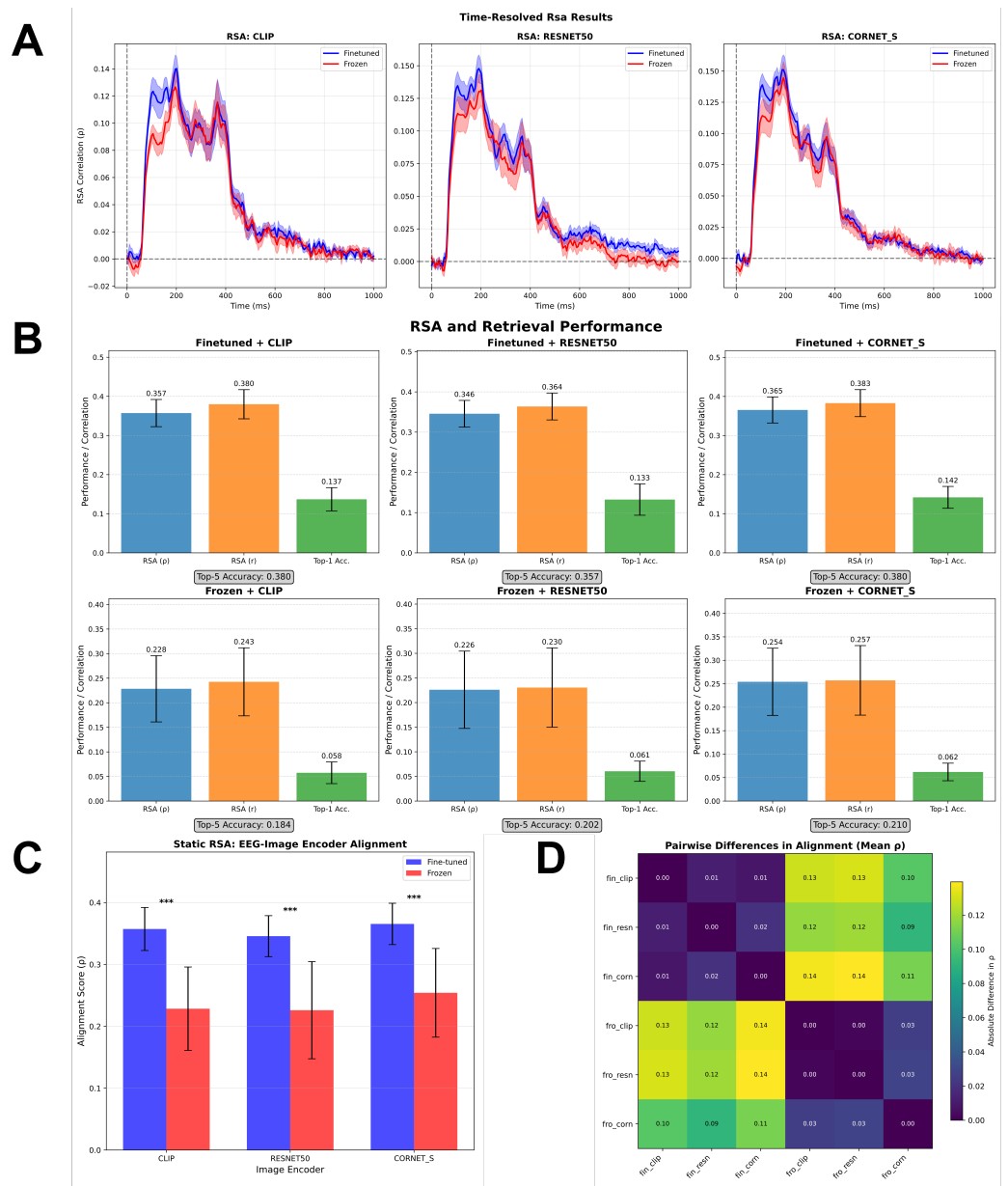

Figure 7: Results of representational analyses. (A) RSA correlation of raw EEG signals with image representations using time-resolved RSA analysis. (B) Mean Pearson ($\rho$) and Spearman ($r$) coefficients for RSA between EEG and image representations for all subjects, along with top-1 and top-5 EEG-to-image retrieval accuracies across model configurations. (C) Comparison of EEG-Image representation alignment between fine-tuned and frozen paradigms using RSA between EEG and Image representations averaged over all subjects (*** indicates statistical significance of $p < 0.001$). (D) Heatmap of pairwise differences in RSA alignment across all model configurations.

## E    RESULTS OF PRIMARY REPRESENTATIONAL ANALYSES

Figure 7 shows the results of various representational analyses.

### E.1 Analysis of temporal dynamics in raw EEG data

The first set of analyses evaluated the extent to which the learned image representations in the shared space captured the temporal dynamics of the raw neural signals. Figure 3 (time-resolved encoding) and 7A (time-resolved RSA) show that the ability to predict or correlate with the raw EEG signal peaks between 100-250 ms and remains significant until around 600 ms post-stimulus. This temporal profile is highly consistent with the known hierarchical progression of feedforward processing along the human ventral visual stream (DiCarlo & Cox, 2007).

Notably, the performance between the fine-tuned and frozen model paradigms is largely comparable in these analyses. This finding is significant: it suggests that the large-scale pre-training of the CBraMod foundation model is sufficient to learn and preserve the core, low-level temporal dynamics of visual neural processing. This supports the use of the foundation model as a strong starting point, as it provides a robust neuro-temporal prior before any task-specific adaptation occurs.

### E.2 Analysis of the aligned shared representation space

The second set of analyses assessed a different, more central question: the quality of the final, shared representational space created by the contrastive learning process. Instead of comparing to raw EEG, these analyses directly measure the geometric alignment between the final EEG representations and the image representations.

The results, shown in Figures 7B, 7C, and 7D, provide consistent evidence for our central hypothesis. The representational alignment, as measured by RSA correlation, is significantly higher in the fine-tuned paradigm compared to the frozen paradigm (Figure 7C, p<0.001). This suggests that while the frozen backbone provides a strong temporal prior, it is insufficient for creating a high-fidelity shared semantic space. The evidence suggests that the process of fine-tuning is important; it allows the model to adapt the general-purpose neural features into representations that are specifically and geometrically aligned with their visual counterparts. The higher correlation values and cross-modal retrieval accuracies (Figure 7B) for the fine-tuned models further confirm the overall effectiveness of the BrainAlign framework in learning a robust, bidirectionally useful shared space.

## F Details of additional representational and semantic analyses

This section provides a detailed description of the methods, experimental setup, and expanded discussion for the additional representational analyses presented in Section 4.4.

### F.1 Methodology

#### F.1.1 Linear isolation of subject-agnostic components (CCA-INLP)

To isolate stimulus-driven information, we first standardize train-time EEG and image embeddings (from a 9-subject pool in a LOSO setup) and compute a $q$-dimensional shared latent space via canonical correlation analysis (CCA (Hotelling, 1992); some modern multiview uses e.g., Andrew et al. (2013)). We report the canonical correlation spectrum as a stability diagnostic. Then, within this EEG CCA space, we use iterative nullspace projection (INLP) (Ravfogel et al., 2020). We train a linear multinomial probe to predict subject identity, compute the probe's row-space, and project the features onto its orthogonal nullspace. This process is iterated until the validation subject-ID accuracy approaches chance, thereby removing all linearly decodable subject information while preserving directions not used by the subject classifier.

#### F.1.2 Mean-subspace removal baseline

As a strong linear control, we compute the between-subject scatter matrix $S_b$ from the class (subject) means on the training EEG features. We then project out the top-$r$ eigenvectors of this matrix (where rank $\leq$ #subjects$-1$). This procedure removes mean or batch-like effects but leaves within-class covariance differences intact, which is conceptually related to second-order alignment methods like CORAL (Sun & Saenko, 2016).

### F.1.3 SEMANTIC SIMILARITY AND RELIABILITY METRICS

To quantify the geometric and semantic structure of the representation space, we compute several metrics from the EEG-to-image cosine similarity matrices (both per-subject and averaged across subjects). These include: (i) Mean Reciprocal Rank (MRR); (ii) category-level Normalized Discounted Cumulative Gain (NDCG@K), which uses graded relevance for same-category items (Järvelin & Kekäläinen, 2002); (iii) ROC-AUC for same vs. different category discrimination (Fawcett, 2006); (iv) within–between category margins and Cohen's $d$ to measure block coherence; (v) a block-energy ratio (the fraction of similarity mass within category blocks); (vi) per-category margins; and (vii) category-centroid consistency across subjects (pairwise cosine similarity), which is conceptually linked to hyperalignment and the analysis of common representational spaces (Haxby et al., 2011; Kriegeskorte et al., 2008).

### F.2 EXPERIMENTAL SETUP

The methods described above were applied in a leave-one-subject-out (LOSO) evaluation framework. In each fold, 9 subjects form the train/validation pool, with the remaining subject held out for testing. All EEG and image embeddings are generated by subject-specific fine-tuned contrastive models (the CORNet-S variant). Test-time performance is measured as 200-way zero-shot identification from EEG to images. For all categorical analyses, we used the 27 WordNet-derived categories provided in the THINGS-EEG2 dataset, of which 16 were present in the held-out test data. We report subject leakage (multinomial probe accuracy on the 9-subject validation pool) and zero-shot top-1/top-5 accuracy on the held-out subject to evaluate the effectiveness of the subject-identity removal techniques.

### F.3 DISCUSSION OF RESULTS

### F.3.1 SUBJECT IDENTITY REMOVAL WITHOUT TASK LOSS

The first ten canonical correlations are high and tight across folds, indicating a stable shared latent despite subject-specific heads (Table 3). As summarized in Table 4, the CCA-INLP pipeline successfully reduces subject-ID leakage to chance while keeping zero-shot recognition unchanged. In contrast, removing only the mean subspace reduces but does not eliminate leakage, suggesting identity is not merely a mean or second-order effect. The fact that a compact linear subspace carries subject identity is a key finding; removing it via INLP drives leakage to chance without degrading recognition, implying the preserved dimensions are predominantly subject-agnostic (stimulus-driven). The insufficiency of mean-subspace removal indicates the presence of residual, higher-order subject-dependent structure.

### F.3.2 SEMANTIC STRUCTURE AND CROSS-SUBJECT RELIABILITY

The subject-averaged space also exhibits non-trivial semantic organization. The retrieval metrics in Table 5 show that correct matches and same-category items rank near the top (MRR/NDCG), and within-category similarity exceeds between-category similarity (positive $\Delta$ and small but non-zero $d$). As seen in Table 7 and the heatmap in Figure 5A, we observe non-trivial block structures for categories like weapon, plant, vehicle, and furniture, indicating robust shared semantic representation. The qualitative retrieval examples in Figure 5B support this, showing that retrieved images are mostly from semantically similar categories. However, for some under-represented categories, retrieval appears to be based on lower-level features like color and shape patterns rather than pure semantics. Overall, the modest global AUC and moderate centroid consistency (with high per-image rank variance) reveal residual cross-subject idiosyncrasies, which is consistent with a shared but imperfectly aligned semantic geometry.

## G ADDITIONAL RESULTS

Tables 8 and 9 present additional top-5 accuracy results for the EEG-to-image and image-to-EEG 200-way zero shot classification tasks respectively.

Table 7: Per-category within-between margins: $\mu_W$ is within mean, $\mu_B$ is between mean, and $\Delta$ is within–between delta. Positive $\Delta$ indicates clearer category blocks in the averaged similarity.

| Category | $\mu_W$ (within) | $\mu_B$ (between) | $\Delta$ (W-B) |
|---|---|---|---|
| animal | 0.0238 | 0.0048 | 0.0191 |
| clothing | 0.0934 | 0.0029 | 0.0905 |
| container | 0.0448 | 0.0091 | 0.0357 |
| dessert | 0.0399 | 0.0094 | 0.0304 |
| food | 0.0554 | 0.0076 | 0.0478 |
| fruit | 0.0592 | 0.0140 | 0.0452 |
| furniture | 0.0907 | 0.0102 | 0.0805 |
| musical instrument | 0.0523 | 0.0112 | 0.0411 |
| plant | 0.1250 | 0.0060 | 0.1190 |
| sports equipment | 0.0959 | 0.0217 | 0.0742 |
| tool | 0.0869 | 0.0213 | 0.0657 |
| toy | 0.0914 | 0.0203 | 0.0711 |
| vegetable | 0.0092 | 0.0095 | -0.0003 |
| vehicle | 0.0955 | 0.0098 | 0.0858 |
| weapon | 0.1520 | 0.0174 | 0.1345 |

Table 8: A comparison of different model performances (top-5 accuracies) across 10 subjects for the EEG-to-image 200-way zero-shot classification task

| Method | S1 | S2 | S3 | S4 | S5 | S6 | S7 | S8 | S9 | S10 | Mean | SD |
|---|---|---|---|---|---|---|---|---|---|---|---|---|
| BraVL(Du et al., 2023) | 17.9 | 14.9 | 17.4 | 15.1 | 13.4 | 18.2 | 20.4 | 23.7 | 14.0 | 19.7 | 17.5 | 3.2 |
| NICE(Song et al., 2023) | 36.6 | 33.9 | 39.0 | 47.0 | 26.9 | 40.6 | 42.1 | 49.9 | 37.1 | 41.9 | 39.5 | 6.5 |
| NICE-GA(Song et al., 2023) | 40.1 | 40.1 | 42.7 | 48.9 | 29.7 | 44.4 | 43.1 | 52.1 | 39.7 | 46.7 | 42.8 | 6.1 |
| CBraMod (fine-tuned) + CLIP | 37.0 | 30.5 | 37.0 | 31.0 | 29.5 | 49.5 | 36.0 | 44.0 | 39.0 | 46.5 | 38.0 | 6.9 |
| CBraMod (fine-tuned) + ResNet-50 | 29.0 | 34.0 | 34.0 | 29.0 | 30.5 | 52.0 | 29.0 | 47.0 | 31.5 | 41.0 | 35.7 | 8.2 |
| CBraMod (fine-tuned) + CORNet-S | 31.0 | 39.0 | 36.0 | 40.5 | 24.5 | 50.5 | 37.5 | 41.5 | 32.0 | 47.0 | 37.9 | 7.7 |
| CBraMod (frozen) + CLIP | 12.5 | 16.5 | 19.0 | 24.5 | 13.0 | 22.5 | 14.5 | 22.0 | 12.5 | 27.0 | 18.4 | 5.4 |
| CBraMod (frozen) + ResNet-50 | 18.0 | 17.0 | 18.5 | 20.0 | 18.5 | 29.5 | 17.0 | 26.0 | 15.0 | 23.0 | 20.2 | 4.5 |
| CBraMod (frozen) + CORNet-S | 17.0 | 22.0 | 24.5 | 25.0 | 21.0 | 25.0 | 18.5 | 23.0 | 12.0 | 22.0 | 21.0 | 4.1 |

Table 9: A comparison of different model performances (top-5 accuracies) across 10 subjects for the image-to-EEG 200-way zero-shot classification task

| Method | S1 | S2 | S3 | S4 | S5 | S6 | S7 | S8 | S9 | S10 | Mean | SD |
|---|---|---|---|---|---|---|---|---|---|---|---|---|
| CBraMod (fine-tuned) + CLIP | 54.0 | 45.5 | 50.5 | 55.5 | 45.0 | 58.0 | 51.0 | 58.5 | 51.5 | 60.5 | 53.0 | 5.3 |
| CBraMod (fine-tuned) + ResNet-50 | 42.5 | 54.5 | 48.5 | 47.0 | 47.0 | 60.5 | 47.0 | 58.5 | 45.0 | 55.0 | 50.5 | 6.1 |
| CBraMod (fine-tuned) + CORNet-S | 49.0 | 57.5 | 47.5 | 53.0 | 43.0 | 67.0 | 52.5 | 64.5 | 49.5 | 63.5 | 54.7 | 8.1 |
| CBraMod (frozen) + CLIP | 15.0 | 25.5 | 25.5 | 33.5 | 24.5 | 28.5 | 17.5 | 31.0 | 18.5 | 30.5 | 25.0 | 6.2 |
| CBraMod (frozen) + ResNet-50 | 17.5 | 25.0 | 21.0 | 27.5 | 27.5 | 37.5 | 20.5 | 37.5 | 18.5 | 32.0 | 26.4 | 7.4 |
| CBraMod (frozen) + CORNet-S | 20.0 | 30.5 | 27.5 | 32.5 | 27.0 | 28.5 | 24.5 | 34.5 | 15.5 | 36.0 | 27.6 | 6.4 |

# H    LLM USAGE

We used large language models to polish writing. All LLM outputs were manually verified and edited by the authors before inclusion.

