# OpenReview forum: "BrainAlign: Leveraging EEG Foundation Models for Symmetric, Interpretable Alignment with Visual Representations"
_ICLR.cc/2026/Conference — ICLR 2026 Conference Withdrawn Submission_

### Official Review · Reviewer_xg4X · 2025-10-30

**Soundness:** 3
**Presentation:** 3
**Contribution:** 2
**Rating:** 2
**Confidence:** 4

**Summary:**

The paper builds upon a large-scale pretrained EEG foundation model (CBraMod) to learn brain-aligned representations. The proposed method
BrainAlign is  a contrastive learning framework that uses a projection network to align EEG features with those from image encoders.  Improved accuracy over the existing methods is presented as well as a reduced training time.

**Strengths:**

The paper integrates existing techniques and addresses training efficiency in aligning EEG features with those of image encoders. the paper addresses relevant problem and provides a meaningful discussion on how the presented work relates to the state of the art. The paper is clearly written and experimental evaluation supports claims on approved accuracy.

**Weaknesses:**

The paper states to be biologically inspired but fails to give a detailed explanation of what exactly is biologically inspired. It claims robust performance, but given that the focus is on subject-dependent models, robustness could have be explained better.  I understand the robustness from the perspective of individual studies but how the robustness is actually tested even on these is not well described. The results are on-pair or better than comparable  approaches but the training converges after 60 epochs and 70%decrease in training time is reported.

**Questions:**

What is the evidence of the approach being biologically inspired?
How is robustness of the method tested?

---

> ### Author Response · Authors · 2025-11-23
> **Rebuttal by the authors**
>
> Thank you for the review!
>
> Following are the responses to the questions raised:
>
> **Clarification with respect to biological inspiration:** The main evidence of biological inspiration is in the design of the custom projection network designed using well-known neuroscientific evidence on visual information processing (section 3.1, lines 161, 182-185). Visual information processing, with respect to object identity, occurs through the ventral “what” pathway. By incorporating learnable weights for different brain regions (primarily divided keeping the visual pathway in mind), we provide the model a prior on the information grouping and weighting. Later visualization of the learned weights provides evidence that the learned weights closely resemble the information weightage in the ventral visual pathway (with primary visual processing happening in early areas). We are working on the latest revision, and we will include these details in the revised draft.
>
> **Clarification regarding robustness of the method:** In the paper, the CCA-INLP analyses (presented in section 4.4 and Appendix section F.3.1) confirmed that there existed a subject-agnostic and stimulus-driven representation space, and that the subject-specific features existed in a small subspace, which could be removed without affecting the overall performance of the model, thus providing subject-wise robustness. This is also loosely confirmed by the relatively lower standard deviations reported in Table 1.
>
> We would be happy to address any other concerns that you might have.

---

### Official Review · Reviewer_u5Gt · 2025-10-30

**Soundness:** 3
**Presentation:** 3
**Contribution:** 2
**Rating:** 4
**Confidence:** 2

**Summary:**

In this paper the authors contribute BrainAlign, a framework to learn a shared representation space of image and EEG signals, leveraging pretrained image encoders, EEG foundation models, and trainable projection modules to the common latent space. The authors evaluate BrainAlign across cross-modal, zero shot classification tasks, highlighting that it outperforms standard multimodal approaches (BraVL, NICE, NICE-GA), requiring less training time to achieve competitive results. Furthermore, the authors present extensive evaluations on the interpretability of the model and the quality of the learned representations, highlighting the connections to visual processing in the brain and the bidirectionally of the framework. Finally, the authors discuss the extent of subject-specific and subject agnostic information within the learned representation.

**Strengths:**

- Overall, the work presented in this papers is quite substantial, and clearly shows the effort done by the authors. In particular, I thoroughly enjoyed the substantial (and insightful) evaluations over the learned representations, present both in the main paper and Appendix.

- The idea of employing EEG foundation models for image retrieval tasks, instead of training from scratch, appears to be novel (to the best of my knowledge). Moreover, the use of a self-supervised loss for aligning the image and EEG representations is a sound methodology. The bidirectionality of the framework is also a nice bonus, yet I believe it to be a byproduct of the self-supervised loss employed, and not particularly novel (see, for example, [1]).

- I believe this paper to be of some significance to the community. The successful demonstration of the use of EEG foundation models for image retrieval tasks, as shown in this paper, can lead to the development of the cross-modal capabilities of these models to other modalities (such as sound, video). Furthermore, the discussion over the properties of the multimodal representation learned by BrainAlign can lead to the development of improved fine-tuning techniques of these foundation models, that target participant-specific information.

- Finally, the paper is well-written, without any major typo. The images are also of high-quality (yet could be slightly bigger to facilitate the interpretation). The document is also well structured, one exception being Section 4.4 and Section 4.5. which could benefit from a reformulation (see weaknesses below).

**References**:
- [1] Rajabi, Nona, et al. "Human-Aligned Image Models Improve Visual Decoding from the Brain." Forty-second International Conference on Machine Learning. 2025.

**Weaknesses:**

- While the presented framework is sound, it does suffer from a lack of novelty: the use of a self-supervised loss to align EEG signals and image representations encoded from pretrained visual encoders has been extensively demonstrated in literature before (see [1,2], as well as other references discussed by the authors in Section 2). The novelty of the work lies in the replacement of the fully-trainable EEG encoder with a pre-trained EEG encoder. However, to achieve reasonable performance, the pre-trained EEG also requires fine-tuning. While the authors highlight that this leads to a "70% reduction in training time (Line 316)", EEG encoders are usually of a much smaller complexity (e.g., in terms of parameters) than the image encoders, and their training time is usually (comparitavely) neglectable. What could be more interesting is, instead, to evaluate: (i) how much data does the fine-tuning process require against the training from scratch procedure? (ii) how much of the original encoder can remain frozen, and how much it needs to be fine-tuned?

- Another apparent disadvantage of the model is that the best performing framework proposed by the authors still underperforms in EEG-to-image retrieval against the NICE-GA (as shown in Table 1). That is not necessarily a problem, if the authors show that their approach improves other measures in comparison with NICE-GA. However, NICE-GA is not a baseline comparison of the extensive evaluations present in Sections 4.3 and 4.4, so it is not clear how they compare.

- While the extensive evaluations of the learned representations in Section 4.3 and 4.4. are welcomed and insightful, the presentation of these results do a disservice to the work shown. I would recommend the authors to unify and streamline their evaluation section, clearly highlighting the main takeaways of the different experiments. Currently, Section 4.4. appears to contain extra irrelevant experiments (the title of the subsection is "Additional Analyses"), which is obviously not the case.

- Some of the Figures could be substantially improved: Figure 4 does not include a legend to identify what are each of the lines, what is the colored area (std, 95% CI?), etc... Figure 5 is also too small, making the text unreadable, which is surprising since the authors still have available space in the document.

- I also found it disappointing that the main paper does not include a dedicated section to discuss the ethical considerations of this work. While the use of this technology in the future can have benefits for certain populations, it's also obvious that it can be used for nefarious purposes, and this should be discussed in the context of the paper.

**References**:
- [1] Song, Yonghao, et al. "Decoding natural images from eeg for object recognition." arXiv preprint arXiv:2308.13234 (2023).
- [2] Rajabi, Nona, et al. "Human-Aligned Image Models Improve Visual Decoding from the Brain." Forty-second International Conference on Machine Learning. 2025.

**Questions:**

- Can the authors present the ablation study described in the first point of the weaknesses? Also, can the authors present the full training time comparisons of the different models to substantiate the statement in Line 316?

- Can the authors compare their approach to NICE-GA in the tasks presented in Sections 4.3 and 4.4?

- In Figure 2 the authors show that the fine-tuning procedure is fundamental to achieve a more biological plausible distribution of weights across the different brain regions. Why this difference? Wouldn't the pretrained EEG model also have learnt a biological plausible distribution, from the large-scale EEG datasets it was trained on? I understand there are subject-specific changes, but wouldn't the "average distribution" learned by the foundation model also produce a similar distribution to the fine-tuned one presented in Figure 2?

- While it is understandable the differences in the results of Figure 3, it would be clarifying if the authors could elaborate on the connection between these differences and the temporal processing of visual information in the brain. Currently there is no explanation in the main text, beyond the statement: "our aligned representations captured significant, dynamically evolving neural information, mirroring the know temporal progression of the visual" (Line 371). Can the authors elaborate on this?

---

> ### Author Response · Authors · 2025-11-23
> **Rebuttal by the authors**
>
> Thank you for the insightful review!
>
> Following are the responses to each of the questions raised:
> 1. The first point of the weakness section is a valid point, and we thank the reviewer for raising the same. We are working on different data splits to address the points of ablation mentioned, and we are working towards including this in the revised draft. The training time reduction was computed with respect to the number of epochs. The number of epochs used by the baselines was 200, while our method achieved convergence in around 50-60 epochs on the same dataset. The data used for fine-tuning our model was the same as the data used for training the model from scratch for the baselines.
> 2. Thank you for asking this question. Sections 4.3 and 4.4 were provided to intrinsically analyze the representation space and were not intended as a comparative study. Those sections were intended to study the interpretability and the information content of the representation space.
> 3. **Clarification regarding learned distribution of weights:** As clearly shown in Figure 2, the model with the frozen pretrained encoders did learn a potentially biologically plausible weight distribution, but as we can see, the weight distribution is more uniform across all regions except the occipito-parietal regions (which indicates earlier visual processing). But after fine-tuning, the weights became more focused in regions associated directly with the ventral visual stream (including both occipito-parietal and temporal regions), thus providing evidence in support of the fundamental role of fine-tuning. If there are any other concerns, we would be happy to address them.
> 4. **Clarification regarding Figure 3 results:** The 3 plots in Figure 3 represent prediction of raw EEG signals from image representations. In early visual processing (the parts from 0 ms to around 400 ms), the brain actively processes the image without much contextual information. For example, there are well-known P100 (the first spike in all 3 plots) and N170 ERPs which result in relatively high signal-to-noise ratios. During this early time, the brain likely processes edges, textures, colors, and other image-specific information, due to which, prediction of these signals from image representations in this time frame becomes relatively easy. On the other hand, in the later stages, the brain transitions from active processing of image features (which are encoded in our image representations) to passive contextual processing not directly related to image features. Due to this transition, direct prediction of EEG from image representations becomes more difficult, since the brain effectively moves out of that feature-processing stage. We are working on the revised draft, and we will be adding a small appendix section to the latest revision to include this information in the draft. If there are any other concerns, we would be happy to address them.
>
> Some comments on the weaknesses cited:
>
> We thank the reviewer for their suggestions. We are currently working on the latest revision and will include the suggested revisions for sections 4.3 and 4.4, along with a dedicated section for ethical considerations.
>
> **Clarification regarding Figure 4 and Figure 5:** We will be updating the Figure 4 caption to indicate the requested information in the revised draft. Each colored line represents one subject, while the shaded area represents the 95% CI. Originally, after increasing the size of Figure 5, the overall size of the draft was overflowing, and therefore it was reduced. But after revisions, the overall content was reduced, creating extra space. We are working on the revision and will increase the size of Figure 5 to enhance its readability in the revised draft.
>
> **Clarification regarding performance comparison to baselines:** As explained in response to another review, the main advantage of the method is the utilization of pre-trained models for achieving competitive performance. There are a few points which strengthen the validity and applicability of our proposed method. First is that the performance of our proposed method is better as compared to the NICE baseline, while remaining competitive to the NICE-GA baseline, while reducing convergence time. Second, the usage of a pre-trained EEG encoder instead of a custom-designed one potentially enhances the transferability of our proposed method to multiple cross-modal tasks involving EEG signals, since the backbone in our method has been pre-trained on a large-scale EEG dataset with over 27000 hours of data, thus providing a strong inductive bias to the entire model. Third, the proposed method is inherently designed to have enhanced interpretability as a consequence of the custom projection network. Finally, the extensive representation analysis results provide evidence towards a high-quality, robust, largely subject-agnostic, and bi-directional learned representation space (as opposed to a unidirectional, task-specific space).

---

> > ### Comment · Reviewer_u5Gt · 2025-11-28
> > **Response to author's rebuttal**
> >
> > I thank the authors for their rebuttal. For now I will maintain my score, but I am open to increasing the score if the authors provide concrete evidence (e.g., through an updated version of the document) of the changes.

---

### Official Review · Reviewer_vcnV · 2025-10-31

**Soundness:** 3
**Presentation:** 3
**Contribution:** 2
**Rating:** 4
**Confidence:** 5

**Summary:**

This paper used pretrained EEG foundation model to enhance the constrastive alignment between visual stimuli and brain responses for visual decoding. It gave interesting attempts to finetune pretrained model for other tasks. Apart from the decoding performance, they claimed that the training time was significantly reduced by 70% with the help of pretrained EEG encoder. The demonstrated the performance with eeg-to-image retrieval and also image-to-eeg retreival, and observed notable correlation with neuro-inspired image model.

**Strengths:**

1. It's a foundamental question that how to use pretrained EEG model in down-stream task. This paper focues on the problem and gave a good evaluation in the visual decoding with contrastive learning framework.
2. The set bidirectional task to test the representation provided by finetuned foundation model.
3. They gave clear results comparison to show the performance of the model without a overly fit results.

**Weaknesses:**

1. It's not clear how the work applied finetuning of the CBraMod and the specific computation cost reduction with the finetuning compared to train a new model.
2. It would be benifit to give a clear description on how to implement EEG-to-image and image-to-EEG tasks, as well as the meaning of the visualization (Fig. 2 and 3). It's a little blurred that what visual information obtained by the model.
3. A summary of the improvement taken by including the pretrained model would help to figure the main contribution of the work. Would that help achieve better prediction results, lower computational cost, clearer brain pattern extraction?

**Questions:**

1. Would different finetunes have impact on the overall performance?
2. How did you use CBraMod here adjusting to the channel settings of the dataset used in this work?
3. What concolsion was give by the CCA-INLP analysis?
4. I still curious about the brain patterns achiveved by the framework, including frozen and finetuned CBraMod model.

---

> ### Author Response · Authors · 2025-11-23
> **Rebuttal by the authors**
>
> Thank you for your review!
>
> Following are the responses to each of the questions raised:
> 1. Thank you for raising this point. This is a valid point and would potentially require a detailed comparative study for testing different fine-tuning strategies and designs. While the current revision does not directly address this, as the central aim was just to show that basic fine-tuning already achieves competitive performance with interpretable modeling, we are actively working on including the comparative study in the revised draft.
> 2. **Clarification about CBraMod usage:** CBraMod, and many modern EEG foundation models, are designed to be channel-agnostic, and allow the processing of data in different EEG formats. The only main requirement to directly use the pre-trained weights is the sampling frequency, and this has been clarified on line 677. After applying basic pre-processing, the fine-tuning was performed simply by starting with the pre-trained weights and training using the dataset mentioned in the paper. One additional benefit of using this particular dataset was that the EEG recording was done using a standard electrode system (10-10), which was easily adaptable to the format required by CBraMod, since the pre-training was done using 10-20 system, all of which exist in the 10-10 system. We just selected the right electrode positions which perfectly matched with the pre-training setup of CBraMod. Furthermore, we have cited the original CBraMod paper in the main text, which can provide the detailed method used to provide channel-agnostic adaptation.
> 3. **Clarification about conclusions from CCA-INLP:** The conclusions obtained from the CCA-INLP analysis have been included in section 4.4 between lines 403 and 409, and in Appendix section F.3. We are happy to resolve any further queries with respect to this.
> 4. Section 4.3 exclusively tackles the question of interpretability of the learned representations (brain patterns). In particular, figures 2 and 3 provide some observations which form the relation between EEG patterns and the learned representations. Additionally, Appendixes D through F provide details of the representational analyses performed to probe the learned representation space, along with some interpretations related to the properties of this learned space. We kindly request the reviewer to refer to these sections and figures. We are happy to address any other queries related to this.
>
> Some comments on the highlighted weaknesses:
> 1. The finetuning process has been clarified in the points provided above. As for the computation cost reduction, the models trained from scratch using the NICE framework (the SOTA) required 200 epochs for convergence, while the fine-tuned models only took 60 epochs, which was a 70 % reduction, as mentioned in lines 315-316.
> 2. The method is outlined in section 3.2. Similar implementation was used during evaluation for EEG-to-image and image-to-EEG tasks.
> 3. Table 1 provides the results comparing the usage of pretrained models with other baselines. As explained in section 4.2 (lines 312-319), usage of pretrained model helps in achieving competitive performance against models trained completely from scratch (which require more time and resources). Therefore, usage of pretrained models, according to our experiments, reduces computation cost, provides a prior on the representation space, and achieves competitive performance. Further experiments using pretrained models could provide even better results by experimenting with post-training techniques and augmentations to the model architecture (like projection modules).

---

### Official Review · Reviewer_edLP · 2025-11-01

**Soundness:** 2
**Presentation:** 2
**Contribution:** 2
**Rating:** 4
**Confidence:** 2

**Summary:**

The paper proposes BrainAlign, a contrastive learning framework that leverages a pretrained EEG foundation model to align EEG features with visual representations from various image encoders . The authors claim that their model achieves biologically plausible alignment, symmetric bidirectional mappings (EEG↔image), and competitive results on the THINGS-EEG2 dataset with reduced training cost.

**Strengths:**

The paper addresses an important problem: improving EEG-vision alignment using foundation models.

**Weaknesses:**

1. The argument for biological plausibility is rather vague. The paper states that fine-tuned models show "biologically plausible attention patterns”. However, I would say these observations should be framed as interpretability, which is different from biological plausibility. The same for the use of the “mechanistic interpretability”  term.
2.  For the EEG to image direction, the model’s advantage over state-of-the-art methods is not clearly established. A proper statistical test (e.g., Wilcoxon signed-rank) should be reported for these comparisons, not only for fine-tuning vs. frozen variants but also versus existing baselines. The small improvement may not justify fine-tuning a model per subject unless there is clear evidence of superior performance or interpretability.
3. There is no comparison with other models for the image to the EEG direction. Although the authors acknowledge this in their limitations section, the subject-dependent setup is a major limitation.
4. While fine-tuning improves performance, the gain is modest.
5 It is unclear whether the results presented, particularly in Tables 1 and 2, are based on a single run or averaged across multiple random seeds.

**Questions:**

See weaknesses

---

> ### Author Response · Authors · 2025-11-23
> **Rebuttal by the authors**
>
> Thank you very much for your insights!
>
> Following are the responses to each of the questions raised:
>
> 1. We apologize for any confusion caused. Our primary aim, through those statements, was to state that the observations point in the direction of potential biological plausibility, rather than definitively proving biological plausibility. For example, the statements on lines 364 and 366 mention biological plausibility as a potential consequence, but not a definite one. The aim of all of those statements was primarily in the direction of enhanced interpretability. As such, we are working on the draft revision, which will contain rephrasing of multiple statements which originally could have caused this confusion. Additionally, as a note, section 4.3 was meant for probing the interpretability of the model, rather than proving the biological plausibility of the model.
> 2. We thank you for pointing this out and confirm that we understand that such a comparison should have been reported in any general case. But it deviates from the aim of this particular study. This paper claims that at least comparable performance can be achieved by using pre-trained encoders (at a lower training cost) instead of designing custom encoders for every task and pre-training and fine-tuning them from scratch. Additionally, using per-subject models was a design choice which allowed learning some subject-specific nuances for more personalized models, rather than learning only average representations for a population, which can potentially be harder to actually apply in real life.
> 3. While this is a valid point, Table 2 was simply included to demonstrate the cross-modal capability of the proposed model and was intended as an additional analysis of a potential capability, and not related to the primary claims, due to which baseline comparisons were not included. We apologize for not making this clear in the main text. This will be addressed in the draft revision that we are working on. For the second concern, as you rightly said, this has been acknowledged in our limitations section, and as explained in the previous point, it was a specific design choice rather than a direct limitation of the method. The same experiments can optionally be carried out for subject-agnostic models, but that would have deviated from our primary aim.
> 4. As explained above, while the gains seem modest, the main advantage of the method is the utilization of pre-trained models for achieving competitive performance. Considering future applications, designing and training complex models from scratch for each application would require a large number of resources, and replacing such models with large-scale pre-trained models could provide a cheaper, yet highly effective and generalizable alternative for deployment. Moreover, there are a few other points which strengthen the validity and applicability of our proposed method. First is that the performance of our proposed method is better as compared to the NICE baseline and competitive to the NICE-GA baseline, while reducing convergence time. Second, the usage of a pre-trained EEG encoder instead of a custom-designed one potentially enhances the transferability of our proposed method to multiple cross-modal tasks involving EEG signals, since the backbone in our method has been pre-trained on a large-scale EEG dataset with over 27000 hours of data, thus providing a strong inductive bias to the entire model. Third, the proposed method is inherently designed to have enhanced interpretability as a consequence of the custom projection network. Finally, as discussed in sections 4.3 and 4.4, and in Appendixes D, E, and F, the extensive representation analysis results provide evidence towards a high-quality bi-directional learned representation space (as opposed to a unidirectional, task-specific space), which is robust and largely subject-agnostic, proving that the model did not just memorize subject-specific idiosyncrasies, but rather learned a potentially universal representation.
> 5. While the current results have been provided for single runs, with standard deviation and average across subjects, we are working on the draft revision, which will contain results across multiple random seeds.

---

> > ### Comment · Reviewer_edLP · 2025-11-27
> >
> > Thank you for the response. I will maintain my score. Overall I  I feel that my specific concerns have not yet been addressed in a concrete way, with more than one point in the revision referring to ‘we are working on a draft revision’.

---

### Note · Authors · 2025-12-05

I have read and agree with the venue's withdrawal policy on behalf of myself and my co-authors.